# A Systematic Review of the Effects of Caffeine on Basketball Performance Outcomes

**DOI:** 10.3390/biology11010017

**Published:** 2021-12-23

**Authors:** Zhi Sen Tan, Alexiaa Sim, Masato Kawabata, Stephen F. Burns

**Affiliations:** Physical Education and Sports Science, National Institute of Education, Nanyang Technological University, Singapore 637616, Singapore; tanz0113@e.ntu.edu.sg (Z.S.T.); nie184704@e.ntu.edu.sg (A.S.); masato.kawabata@nie.edu.sg (M.K.)

**Keywords:** sports-specific, skill, shooting, dribbling, jump-height, sprints, agility

## Abstract

**Simple Summary:**

Caffeine is a stimulant of the central nervous system widely utilized by many athletes to enhance endurance, strength, and power-based sports performances. Whether ergogenic enhancements following caffeine ingestion result in improvements in sports-specific skills performance has received less attention. In basketball, the ability to execute certain tasks with accuracy, such as shooting and passing, are key factors affecting the outcome of the sport. Besides being able to excel in accuracy-based tasks, the possession of strong physical attributes, including vertical jump height, sprint speed and agility, are also key components to basketball performance. In this review, an overview of the effects of caffeine on basketball-related skill tasks and physical aspects of performance deemed important for the game is provided. One of the key focal points is that the efficacy of caffeine is influenced by a multitude of determinants that have an overall impact on the ergogenic capacity of caffeine. Proper awareness of these determinants allows basketball players, coaches, and trainers to have better insights and knowledge in applying caffeine to improve basketball-related performances.

**Abstract:**

Caffeine is an ergogenic aid in many sports, including basketball. This systematic review examines the effects of caffeine on basketball-related skill tasks along with physical aspects of performance deemed important for the game. A systematic search was conducted across three databases (PubMed, SPORTDiscus and Web of Science) to identify randomized-controlled trials which examined the effect of caffeine on basketball performance outcomes including: free-throw, 3-point shooting accuracy, dribbling speed, vertical jump height, and linear and repeated sprints. Forty-six articles were identified of which 10 met the inclusion criteria. Improvements in vertical jump were identified in four of five studies, agility in two of four studies, and in linear and repeated sprints in two of three studies. No deterioration in basketball skills performance was observed in any studies. It is suggested that caffeine is useful for basketball players to improve the physical aspects of their game-play performance but there is little evidence of any change in skill-based performance at present. Further research should clarify the effects of caffeine on basketball performance in women and the role of individual genetic variation on caffeine metabolism. Basketball players and coaches should be aware of the properties of caffeine before ingesting it as an ergogenic supplement.

## 1. Introduction

Basketball is a demanding sport that requires repeated bouts of movements at a high intensity, such as sprinting, shuffling, and jumping over a prolonged duration [1]. To excel in the game, an elite basketball player would be expected to possess high levels of aerobic and anaerobic capacity, power, speed, strength, and agility [2]. In addition, players must execute skill-related ball-tasks, such as dribbling, passing, or shooting into the hoop. Notational analysis has shown that these skill-related tasks play an important part in determining the outcomes of basketball games [3]. Enhancing the ability to execute these skill-related ball-tasks while simultaneously performing high-intensity movements would thus be essential in increasing the success of the game. Hence, understanding the use of ergogenic aids, such as caffeine on basketball-related performance would be of potential value to basketball players, coaches and trainers involved in the preparation of training programs [2].

Caffeine (1, 3, 7-trimethylxanthine) is an ergogenic aid favored by athletes in a range of sports with research examining its effects on sports/physical performance extending over 100 years [4]. Several previous review papers support the ergogenic effects of caffeine on various types of sports performances, including endurance performance [5], short-term high intensity exercise [6], resistance exercise [7] and in team-based sports [2]. Enhancements in sports performance with caffeine ingestion are attributed to several mechanisms, with the primary suggestion centered on its role as an adenosine competitor [8]. Caffeine diminishes the effects of adenosine on the central nervous system by reducing its ability to downregulate arousal [9] and thereby enhances muscle firing rates [10]. Moreover, caffeine increases the discharge of adrenaline [11] and modifies the metabolic use of fat and carbohydrates in the body [12]. Cumulatively for these reasons, caffeine is the most extensively consumed ergogenic aid by athletes [9].

A previous review by Chia and colleagues [2] explored the effects of caffeine on ball-game performances (invasion games and net-barrier games). It was reported that low to moderate doses of caffeine (3 to 6 mg per kg of body mass [BM]) increased vertical jump and sprint performances in basketball, soccer, rugby, and volleyball simulated matches but there were no significant improvements in total distance covered, agility or accuracy-based performances. However, at that time only two studies on basketball were included in the review (*n* = 21 participants) with the primary focus being physical rather than skill-based performance outcomes. Since that time, considerable information has emerged on how caffeine affects basketball performance which deserves individual attention rather than inferring results from a range of ball-based sports where the key physical and skill demands to winning may be different. A comprehensive review in this area would serve scientists, coaches and athletes exploring the effects of using caffeine for basketball. Thus, the aim of this systematic review is to provide more knowledge and details on the impact of caffeine use on basketball players by exploring the effect of caffeine ingestion on physical- and skill-related basketball performance tasks deemed important for the game: shooting accuracy, passing accuracy, dribbling speed, vertical jump height, change-of-direction agility, and linear and repeated sprints.

## 2. Methodology

A standardized literature search and reporting strategy following the Preferred Reporting Items for Systematic Reviews and Meta-Analyses (PRISMA) was performed for this systematic review (see Appendix A) [13]. The literature search utilized three databases (PubMed, SPORTDiscus and Web of Science) and was performed from January 2001 to November 2021 using the keywords “caffeine” and “basketball”. Manual cross-referencing was conducted on the reference lists of the original research articles to identify additional studies concerning caffeine and basketball. All article titles and abstracts obtained from the search were uploaded to Covidence [14] where duplicate articles were removed. Relevance of articles was then screened based on the article titles and abstracts. The process of identification and assessment of articles was performed by two independent reviewers (TZS and AS), who discussed any discrepancies with a third independent reviewer (SFB) until a consensus was achieved.

The articles were included for review and assessment if they fulfilled these criteria: (1) studies that tested the effect(s) of caffeine on basketball-related performance; (2) included human participants only; (3) included information on the caffeine dose and method of administration; (4) used a randomized, placebo-controlled study design; (5) the articles were available in English; and (6) articles were available as full texts and not abstracts only.

The Cochrane risk-of-bias (RoB 2) tool was used by two independent reviewers (TZS and AS) to assess the risk of bias in evidence obtained from randomized controlled trials. The RoB 2 consists of six different domains of bias with a focus on study design, execution, and data reporting. These six domains include: (1) randomization process; (2) period and carryover effects; (3) deviations from intended intervention; (4) missing outcome data; (5) measurement of the outcome; and (6) selection of the reported result. In each domain, a suggested opinion about the risk of bias was formed using information obtained from the articles to answer several signaling questions. The opinions are divided into three categories: ‘low risk of bias’, ‘some concerns of bias’ or ‘high risk of bias’. After the assessment of risk was obtained from each domain, an overall risk-of-bias judgment was then determined by each independent reviewer.

Secondly, the Physiotherapy Evidence Database (PEDro) scale was utilized independently by each reviewer to assess the quality of the articles. The PEDro scale has two purposes: (1) to assess the internal validity of randomized clinical trials and, (2) to assess if the articles had results that were interpretable based on the statistical information provided. The PEDro scale was important in evaluating the trustworthiness of the treatment effects of caffeine on basketball-related performances. Each article was graded based on a total score of 10 and articles with higher scores correlate with higher quality. Any articles scoring less than six on the PEDro scale were considered low quality and hence, excluded from the review [2]. The independent reviewers then discussed the results of the RoB 2 and PEDro to ensure unanimity.

Extraction of data from included studies was conducted using an adapted version of the “Population, Intervention, Comparison, Outcome” (PICO) framework [15]. The following was obtained from each study: (1) author and year of study, (2) number and profile of participants, (3) study design, (4) caffeine and placebo administration details (dose, form and pre-exercise ingestion time), (5) physiological and skill-related performance outcome measures in the caffeine and placebo interventions, (6) information on objective and subjective physiological measures during caffeine and placebo interventions, (7) data on side effects of the intervention, and (8) significance of outcome measures (*p*-value). Hedges’ *g* was calculated by dividing the mean differences between the caffeine and placebo conditions with the pooled standard deviation. The pooled standard deviation was obtained by dividing the sum of the standard deviations of the caffeine and placebo condition. It was used as an estimate of the effect sizes for the abovementioned pairwise comparisons. The magnitude of the influence of caffeine on each performance measure is based on the absolute value of the effect sizes calculated and interpreted as: trivial (0–0.19), small (0.20–0.49), medium (0.50–0.79) and large (≥0.80) [16].

## 3. Findings

### 3.1. Description of Studies

A preliminary literature search identified 46 articles from which 18 duplicates were removed and a further 18 articles were removed after the screening of the abstract and title (*n* = 15) or the article was not available in English (*n* = 1) or were not randomized-controlled trials (*n* = 2). A total of ten articles were assessed for eligibility and fulfilled all inclusion criteria after quality assessment (Figure 1) [1,17,18,19,20,21,22,23,24,25]. Among the ten articles, two articles [18,19] were identified to have been published from data obtained from the same exercise test session. Results were obtained from 130 participants (range 5 to 21 participants per study). Among the 130 participants, 94 (72.3%) were males and 36 (27.7%) were females. Twenty-seven of the male athletes were categorized as youth athletes (<18 years) while no youth female athletes were recruited in any of the studies. All participants were active athletes who were competing and training regularly in basketball (minimally thrice per week of 90 to 120 min each time) but two studies did not specify the weekly training hours [17,24]. Participants were identified as light caffeine consumers with an average consumption of <200 mg per day in all studies except in the study by Tucker et al. [25] who reported that players’ daily average consumption was <500 mg of caffeine. Five studies included female athletes [18,19,21,23,24] but two reports [18,19] involved the same participants in the same exercise test session. Apart from Tan et al. [24] who did not specify the phase of their menstrual cycle, the remaining studies performed the testing during the luteal phase of the menstrual cycle.

As per the inclusion criteria, all studies were randomized and had a placebo comparison condition. Of the studies reviewed, the common placebos used were cellulose [17,18,19], dextrose [21,22,23], sucrose [20], maltodextrin [24], zero caffeine containing energy drink [1], and vitamin tablets [25]. The dose of the placebos used was likely insufficient to elicit any effect on performance and was hence considered negligible. All studies were double-blinded except for Tan et al. (46) who conducted a single-blind study. Other than three papers [17,21,25] that did not specify the blinding process, the remaining had an investigator uninvolved with the experiment to prepare an alphanumerical code assigned to each trial. Participants were given instructions to standardize their diet and fluid intake in all studies. However, the standardization protocol differed across studies. All but three studies [17,19,24] instructed their participants to abstain from caffeine and alcohol for at least 48 h. Puente et al. [19] encouraged their participants to refrain from consuming caffeine throughout the entire trial while the other two studies had their participants refrain from caffeine and alcohol for at least 24 h before the trials. Participants were also told to refrain from strenuous exercise 24 h prior to attending the trials in four studies [1,19,21,23] to mitigate the effects of any prior muscle soreness or metabolic fatigue.

Based on the RoB 2 assessment, all studies were deemed low risk in all domains with one exception [25] (Figure 2). The study by Tucker and colleagues [25] was deemed to have some concerns of bias as no information was provided on the randomizing process of the intervention order. In addition, multiple t-tests were used instead of an analysis of variance (ANOVA) for statistical analysis. Reviewing the internal validity of the studies using PEDro, all were considered of acceptable quality (with an agreement rate of 100% between each reviewer) with eight having a PEDro score ≥ 9 and two with a PEDro score of 6 [24,25] (Table 1). However, four out of the ten studies [17,21,24,25] did not provide detailed descriptions of the blinding procedure. Additionally, the lower scores allocated to these last two studies were because Tan et al. [24] used a single-blind procedure while Tucker et al. [25] did not provide information on the randomizing process and the effect sizes of the results.

### 3.2. Determinants of Caffeine Efficacy

The dose administered is an important factor in determining the overall ergogenic effect of caffeine [4]. In all studies reviewed, caffeine dose was administered in accordance with the participants’ BM as compared to utilizing an absolute dose. The use of absolute dose may not be ideal due to a multitude of factors affecting the ergogenic effect of caffeine, causing large discrepancies in results [9]. The caffeine dose used in the studies reviewed ranged from 3 to 6 mg per kg of BM, which is accepted as optimal for eliciting ergogenic effects [9]. Of the ten studies, seven utilized a low-dose of caffeine (3 mg per kg of BM) [1,18,19,21,22,23,25], of which six—with the exception of Tucker et al. [25]—found this dose ample enough to improve at least one basketball performance measure. One reason for the exception could be because, in the study by Tucker and colleagues [25], a small sample size was utilized (*n* = 5). The remaining three studies [17,20,24] used a moderate dose of caffeine (6 mg per kg of BM). One study by Tan et al. [24] found no improvement in basketball performance measures while the other two studies reported an improvement in at least one of the performance variables measured [17,20,24].

The timing of caffeine administration is a second factor that may affect its efficacy. Caffeine is rapidly absorbed by the gastrointestinal tract into the circulation upon consumption [2], with plasma concentrations reaching maximal levels approximately 60 min after ingestion [26]. This absorption is unaffected in doses up to 10 mg per kg of BM of caffeine [2,27]. In all the reviewed studies, caffeine was administered to participants 60 min before exercise. However, based on data from studies using cycling as a mode of exercise, it was recently suggested that for doses of caffeine < 2.5 mg per kg of BM, greater performance enhancements are seen when ingestion occurs less than 60 min pre-exercise [9,28]. The authors hypothesized that central and/or peripheral sensitivity to caffeine doses < 2.5 mg per kg of BM may be blunted over time even though plasma concentrations remain elevated and hence, a shorter pre-exercise ingestion time would be more beneficial. Whether a similar situation exists for basketball is uncertain. From the published studies reviewed here, however, none manipulated the timing of ingestion to determine when the optimal effects on basketball performance in relation to caffeine intake occur. Hence, future studies could focus on this aspect of caffeine research.

A final crucial determinant of caffeine efficacy is the mode of administration (e.g., capsule, beverage, drink solution, gum) as this affects the absorption rate into the body [2]. Caffeinated gums result in a quicker absorption of caffeine which may be useful in situations with shorter pre-exercise periods [4,29]. Of the studies reviewed, six administered caffeine via capsule [17,18,19,21,22,23] while two studies administered caffeine by dissolving the powder into a solution [20,24]. One study [1] provided an energy drink that also contained 18.7 mg per kg of BM taurine, 4.7 mg per kg of BM sodium bicarbonate, 1.9 mg per kg of BM L-carnitine and 6.6 mg per kg of BM maltodextrin while Tucker et al. [25] used caffeine tablets containing 10 mg of B1 thiamine for every 100 mg of caffeine. It cannot be ruled out that the use of other active ingredients in a study may blunt or exaggerate the effectiveness of the caffeine intervention in those studies.

### 3.3. Basketball Skills Performance

Table 2 provides information and outcomes on skill-related (shooting and dribbling—please note that no studies reported passing) basketball performance outcomes.

#### 3.3.1. Shooting Accuracy

Notational analysis conducted on basketball competitions has identified that shooting accuracy (2-point, 3-point, and free-throw) is a key indicator of basketball game outcomes [3]. The effect of caffeine on shooting accuracy was reported in three studies [1,19,24], with limited evidence suggesting an improvement in shooting accuracy after ingestion. Furthermore, effect size estimates indicate that any improvements were trivial (Hedge’s *g* trivial to small; range: −0.3 to 0.32). Abian-Vicen and colleagues [1] had participants perform 12 sets of two free-throws with a 30 s rest between sets whereas for Tan et al. [24], the participants carried out five sets of repeated sprints (6 × 15 m) followed immediately by two free-throws and then a 2-min rest before the next set of sprints and free-throws. Puente et al. [19] instructed their participants to complete 10 repetitions of a jump test, agility test and then complete two free-throws. In all three studies, it was found that free-throw accuracy was unaffected by the ingestion of a low to moderate dose (3–6 mg per kg of BM) of caffeine. In the study by Puente and colleagues [19], an additional 20-min, 5 versus 5 simulated game protocol played in two 10-min halves was utilized in 20 professional or semi-professional male and female basketball players (*n* = 10 for each sex). It was found that the number of free-throws attempted and scored in the simulated game was significantly higher in the caffeine than placebo condition. However, one criticism of this study design is that there were differences in the number of free throws taken by each player under the caffeine and placebo conditions because the progression of the game dictated the number of free throws taken [24].

On top of free-throw accuracy, Abian-Vicen et al. [1] used a basketball-specific protocol to investigate 3-point shooting accuracy in 16 junior league national players, which involved shooting up to 21 shots from seven different places around the 3-point line in under 1 min. Puente et al. [19] used the simulated game mentioned previously to analyze both 3-point and 2-point accuracy after caffeine ingestion. In both studies, caffeine administered at 3 mg per kg of BM did not significantly improve 3-point or 2-point shooting accuracy. However, it is worth noting that in the basketball-specific shooting protocol by Abian-Vicen and colleagues [1], there was a slight increase in the number of 3-point shots scored after caffeine ingestion. Potentially, increased arousal from caffeine improved physical performance, in terms of the number of shots taken, without improving shooting accuracy.

The lack of significant improvements in shooting accuracy noted across the three studies here could be due to several reasons. In the study by Abian-Vicen et al. [1], there was no fatigue protocol given to participants prior to shooting the basketball. It was previously shown that fatigue negatively affects the shooting accuracy of young basketball players [30], potentially via a reduced hip joint angle and raised shoulder joint angle when players’ center of mass was at its lowest point [31]. Therefore, the fresh state of the players in the study by Abian-Vicen and colleagues [1] while shooting may have mitigated the ergogenic benefits of the caffeine. For Tan et al. [24], although the participants were required to perform six repeated sprints of 15 m before shooting two free-throws, the fatigue induced may have been insufficient compared with that observed during real basketball games where sprints are performed once every 21 s on average with change of directions accounting for 31% of the movement in games [32]. Finally, the simulated game protocol used by Puente and colleagues [19] was dependent on the progression of the game which suggests that the results could be influenced by other factors like individual skills or psychological ability to perform in a game-like situation. Moreover, the total number of shots for each category of shooting (3-point, 2-point, free-throw) may have been too small to allow accurate/sensitive enough analysis of the effect of caffeine. Thus, at present, there is little evidence to suggest that caffeine affects free-throws, 2-point or 3-point shooting accuracy in basketball. Studies utilizing shooting protocols that implement game-like fatigue are needed in order to obtain more robust conclusions of caffeine’s impact on shooting accuracy under ecologically valid situations.

#### 3.3.2. Dribbling Speed

The ability to dribble the ball while moving at high speeds is an important skill in basketball [21]. A high-tempo offensive strategy, otherwise known as a fast break, in basketball, is considered the quickest and most effective way to transit from defense to offense [33]. Data show that dribbling, rather than passing, is the skill executed in approximately 76% of fast breaks in basketball [21,33].

In the present review, four studies investigated the effect of caffeine on dribbling speeds while performing different types of movements—change of direction, and linear and repeated sprints [19,21,22,23]. No significant effects of caffeine were noted on dribbling speeds in these studies. The effect size estimate suggests that caffeine produces only minimal to trivial improvements in dribbling speeds (Hedge’s *g* trivial to small; range: −0.45 to 0.42). Improvements in dribbling speeds were represented by smaller values of time and hence, denoted by negative effect sizes. The study by Puente and colleagues [19] used the change of direction and acceleration test (CODAT) while dribbling a basketball. This required the participants to complete a course consisting of four directional changes (two 45° turns and two 90° turns) and short distance sprints of 5 m and 10 m each [19]. For linear sprints, the time taken to complete 5 m, 10 m and 20 m sprints while dribbling was recorded in two studies without any improvements noted [21,23]. As such, it appears that pre-exercise caffeine ingestion provides minimal ergogenic enhancement when dribbling at speed regardless of directional changes, linear or repeated sprints.

Interestingly, in the fourth study, Stojanović et al. [22] compared the efficacy of caffeine or placebo administered in the morning with that provided in the afternoon on dribbling whilst performing 140 m suicide runs with several changes of direction involved. Dribbling a basketball after caffeine ingestion was faster in the morning than in the evening with time-of-day and interaction effects detected. Potentially, these data suggest that the use of caffeine as an ergogenic supplement may be more beneficial in the morning than evening. Lower perceived exertion and greater arousal of individuals in the morning or alternatively greater fatigue accumulated by evening testing (tests were run between 21:50 h and 22:30 h) may be an explanation for the difference in caffeine’s observed effects here [22,34]. 

The act of dribbling a basketball is complex and involves multiple technical skills. Its effectiveness is not defined to just being able to move quickly while dribbling the basketball, but current studies have not attempted to include elements beyond change-of-direction while dribbling in current testing protocols. Moreover, upon completion of a dribbling action, subsequent actions like shooting or passing the basketball as well as performing a lay-up could also be said to be affected by the effectiveness of dribbling. Hence, further research under more ecologically valid conditions should be conducted to see if the accuracy of different dribbling skills, as well as the crossover into the subsequent action, are improved by caffeine ingestion.

### 3.4. Physical Performance

Table 3 provides information on physical performance (vertical jump height, sprint times and power output) outcome measures in basketball performance in each study.

#### 3.4.1. Vertical Jump

In many ball games including basketball, the ability to perform a vertical jump is vital for athletes to attain success [2]. Across the studies reviewed, the most common test used to measure the power output of lower extremities as well as the maximal height attained in jumps is the countermovement jump (CMJ) test. The CMJ test requires participants to jump as high as they can from a stationary, vertical position before landing on both feet. Of the four studies in this review that included the CMJ test, three found significant improvement in vertical jump height following caffeine ingestion [1,19,20]. Conversely, Stojanović et al. [23] found no significant improvement in CMJ tests with and without arm swings. Stojanović and colleagues [23] recruited only women while the other studies were performed on men.

The information on whether caffeine exerts differentiating effects on sports performance in men and women is divided [4,35] and it is possible that caffeine’s effects on CMJ in women differ from men (see Appendix B; Table A1). Early reports found that the pharmacokinetics of caffeine in women, who were non-contraceptive users in the follicular phase of the menstrual cycle, were comparable with men both at rest and while exercising [36] (pp. 301–327). Similarly, greater sensitivity to caffeine—particularly in slow twitch fibers—was observed from in vitro muscle biopsies taken from men compared with muscle biopsies from women [37,38]. More recent studies examining sports performance have shown that caffeine has an ergogenic effect during resistance [39] and endurance [40] exercise in women. However, there is greater controversy in studies involving direct comparisons between men and women. Some show comparable ergogenic effects between sexes [41,42] but others have found that caffeine improves sprint power [43], total weight lifted [41] and time to perform a repeated agility test [44] in men more than women. A recent systematic review of 10 studies compared caffeine’s effects on sports performance in men and women. There were no differences between sexes in terms of aerobic performance but there was some indication that anaerobic performances were improved in men compared with women, with men able to produce greater power, lift more weight and improve speed to a greater extent with the same dose of caffeine [35]. Thus, it is possible that caffeine’s effects in a movement, such as CMJ differ between sexes.

We are unaware of any direct data comparing the effects of caffeine performance between men and women on CMJ height. However, Puente and colleagues [19] investigated the effect of caffeine on vertical jump performance using the Abalakov jump protocol in both male and female participants. In the Abalakov jump protocol, the participant starts in a stationary, upright position and then positions his arms above his head, swings his arms down while flexing his knees concurrently before jumping as high as possible and landing on both feet. Whilst it was reported that caffeine ingestion improved Abalakov jump height, gender differences were not examined. Clear data, from carefully controlled studies that control for factors, such as the use of oral contraception and differences in the menstrual stage cycle which can impact the metabolism of caffeine [4], are needed to differentiate the potential effects of caffeine on male and female basketball players.

Interestingly, Stojanović and colleagues [22] examined the effect of time-of-day of caffeine administration on CMJ height and found caffeine to have improved efficacy only in the morning and not the evening. This finding was attributed to a greater ergogenic effect of caffeine on contractile and elastic properties of muscle via improved motor unit recruitment and activation in the morning because vertical jump performance has previously been shown to exhibit a nadir in the morning and peak in the evening [22,35]. From this perspective, the authors suggest that caffeine administration can help nullify differences in neuromuscular performance across the day. For basketball players, this may serve as a method to maximize physical performances in training sessions held at different times of day.

Another test protocol used was the 15 s maximal jumping test (RJ-15) where participants are required to perform maximal, repeated CMJs over a span of 15 s. This protocol may have applicability for basketball where often multiple jumps are performed over a short period of time rather than a single maximal jump [32]. The RJ-15 test was utilized by Abian-Vicen et al. [1] and it was revealed that caffeine ingestion improved average jump height performance. Thus, caffeine ingestion is beneficial for singular and repeated vertical jump performance, which is crucial as it has been reported that basketball players jump an average of 46 times in a real game [32].

Finally, a study by Tucker and colleagues [25] found no significant differences in rebound jump height following caffeine ingestion. With their arms kept by the side, participants were told to jump as high as possible in the quickest time after stepping off a 45 cm platform onto a pressure-sensitive timing mat. Reactive strength index (RSI) was used to evaluate the vertical jump performance and was calculated by dividing the height jumped by the contact time on a pressure sensitive mat. A possible explanation for the inability to find any improvements could be due to the small sample size of this study (*n* = 5) as it was observed that three of the five participants improved RSI in the caffeine condition.

In summary, the majority of evidence favors an effect of caffeine on vertical jump height with effect size estimates implying notable improvements (Hedge’s *g* trivial to large; range: −0.96 to 1.33). Time-of-day may be one factor to consider in the effectiveness of caffeine on this performance measure. More data for the effect of caffeine on this outcome in women is needed.

#### 3.4.2. Agility, Linear and Repeated Sprints

The ability to manipulate one’s bodyweight to allow quick changes in direction is termed as agility. Two test protocols were employed in the studies reviewed to measure agility—the CODAT [19] and the lane agility drill (LAD) [20,22,23]. The LAD requires participants to sprint twice (once clockwise and once counter-clockwise) through a course in the shape of a 5.8 m by 4.9 m square and undertake two lateral side-shuffles in opposite directions as well as one forward and backward sprint. In the study by Puente et al. [19], there were no differences in the completion timing of CODAT. However, significant improvements in the LAD time were found in the studies by Raya-González et al. [20] and Stojanović et al. [22] (only in the morning but not evening trials) while only small non-significant improvements were found in Stojanović et al. [23]. Gender differences in performance enhancement with caffeine may again be a factor for these differences, as the latter study performed the test protocol on female participants only whereas the earlier two recruited only male participants (see Appendix B; Table A1). As with vertical jump, greater focus should be placed on researching the relationship of caffeine with agility performance in female basketball players.

In addition to rapid changes in direction, basketball players are also required to perform high speed sprints on the court. One match analysis on elite women basketball players reported that approximately 57% of sprints were between 1 to 5 m while 30% of sprints were between 6 to 10 m [45]. The papers reviewed here showed mixed outcomes for linear sprint times (Hedge’s *g* trivial to large; range: −1.13 to 0.10). Similar to dribbling speeds, improvements in linear sprint times were represented by smaller values of time and hence, denoted by negative effect sizes. Using a 20 m sprint protocol, Raya-González et al. [20] reported significant improvements in linear sprint times. Similarly, Stojanović et al. [23] found significant improvements in 10 m and 20 m sprint times in the professional female basketball players tested but 5 m sprint times were not significantly improved with caffeine ingestion in the same condition. Conversely, Stojanović et al. [22] found no differences in 5 m, 10 m and 20 m sprint times between caffeine and placebo regardless of the time of day (AM or PM) in national level male youth basketball players. Both studies used the same dose of caffeine (3 mg per kg of BM) and the authors hypothesized that the difference between the two studies might be related to training status with a greater ergogenic effect of caffeine in trained or experienced athletes. In the study by Raya-González et al. [20], they used a higher dose of 6 mg per kg of BM of caffeine. Another possibility could, therefore, be that low dose caffeine improves linear sprint times on females while a higher dose is required for male basketball players.

For repeated sprints, Abian-Vicen et al. [1] used the Yo-Yo Intermittent Recovery Level-1 (IR-1) test, which requires participants to perform 20 m shuttle runs at increasing velocities with an active recovery of 10 s between runs till the participant fails to meet the required completion timing. Although there was no significant improvement following caffeine ingestion, the total distance covered in the Yo-Yo IR-1 test was slightly greater with caffeine. Raya-González et al. [20] administered a repeated sprint test protocol consisting of 5 × 30 m (15 m + 15 m) shuttle sprints with 30 s of recovery between sprints. The total sprint performance and best individual performance were significantly faster after pre-exercise caffeine ingestion. Both studies by Stojanović and colleagues [22,23] used a suicide run, comprising of a 140 m sprint along with numerous changes of direction on the basketball court. The earlier study found no improvements in suicide run times for female basketball players while the latter study reported a significant decrease in suicide run times in the morning but not evening in male youth basketball players with the ingestion of caffeine. The authors suggested that the differential sprint performance elicited was from a greater hypoalgesic effect—as evidenced by a reduced rating of perceived exertion—of caffeine during this type of glycolytic exercise in the morning compared with the evening. Collectively, it can be concluded that pre-exercise caffeine ingestion has an ergogenic effect on repeated sprints for male basketball players (Hedge’s *g* trivial to large; range: −1.15 to 0.11) while further research should focus on female basketball players.

#### 3.4.3. Power Output

Two papers measured the power output of basketball players using different tests with differing results. Abian-Vicen et al. [1] reported an increase in leg muscle power output during the RJ-15 while no improvements were found in mean power during the concentric phase of the jump or peak power output after two maximal CMJ. Potentially, differences in power output during jumping after caffeine may only become obvious in a test where some fatigue occurs, such as the RJ-15 rather than in a single maximal test, such as the CMJ. Cheng et al. [17] measured power output with a 3 min all out cycling test. While mean and peak power output did not improve following pre-exercise caffeine ingestion, the average power output from 0 to 60 s; 0 to 90 s and 0 to 120 s was found to be significantly higher. This suggests that pre-exercise caffeine ingestion could be useful in increasing power output in short durations of 1 to 2 min. However, the relevance and ecological validity of this type of cycling testing for basketball players are questionable.

### 3.5. Physiological and Subjective Responses

Table 4 provides information on physiological changes (heart rate, blood lactate, subjective perceived responses, and body temperature) during basketball testing under caffeine and placebo conditions.

#### 3.5.1. Heart Rate

Heart rate is often used as an indicator for the intensity of sports and exercise [32]. Given the stimulatory effect of caffeine on the central nervous system (CNS), it could logically be assumed that heart rate would be affected by caffeine consumption [2]. Four studies [17,19,24,25] analyzed the effect of pre-exercise caffeine ingestion on heart rate with mixed findings. Cheng et al. [17] reported a higher peak heart rate using a 3 min all out cycling test after 6 mg per kg caffeine ingestion compared with placebo. Similarly, in the study by Tan et al. [24] where participants were required to perform five sets of 6 × 15 m repeated sprints on the basketball court, caffeine increased the mean heart rate in four out of five sets. However, during a 20-min simulated basketball game, Puente and colleagues [19] found no differences in the mean or maximal heart rate between the caffeine and placebo conditions. In the study by Tucker et al. [25], heart rate was analyzed during a graded incremental treadmill test, but no results were reported in the article. Thus, current data suggest that under conditions where the intensity is fixed, higher heart rates may be recorded but under conditions of variable intensity, such as gameplay, heart rate is similar with or without caffeine ingestion. An increased heart rate could be an indicator of fatigue, arousal, or physiological stress during a basketball game. Given the prolonged duration of a basketball game, the ability to manage the fatigue levels or work harder could be beneficial in maintaining peak performance.

#### 3.5.2. Blood Lactate

Blood lactate was measured in two studies [17,25]. In the first, lactate was measured during a 3 min all out cycling test [17] and in the second study during a graded incremental treadmill test [25]. Compared to placebo, Cheng and colleagues [17] observed higher blood lactate levels with caffeine but Tucker et al. [25], did not report the specific blood lactate concentrations. However, it was mentioned that three participants (60%; *n* = 5) experienced raised blood lactate with caffeine. The increased blood lactate observed by Cheng et al. [17] could be due to the increased work performed during the 3 min all out cycling test as power output was greater in the caffeine condition. This suggests that pre-exercise caffeine ingestion may be useful in increasing total work done during exercise. Despite that, the ecological validity of the cycling test and graded incremental treadmill tests used in these studies is questionable in relation to basketball performance.

#### 3.5.3. Self-Perceived Responses

The most common types of self-perceived responses are exertion, performance, and muscle power. Seven out of ten studies measured at least one of the mentioned types of response [1,17,19,22,23,24,25]. Of the seven studies, three [17,24,25] used the 6–20 Borg scale [46] whereas the remaining four studies [1,19,22,23] used a 1–10-point Likert scale to measure perceived exertion. Regardless of scale used, in five studies pre-exercise caffeine did not decrease perceived exertion compared with placebo while performing numerous tests—3 min all out cycling, CMJ, LAD, 5 m sprint, 10 m sprint, 20 m sprint, squat jump, suicide run, graded treadmill test and a simulated basketball game [17,19,22,24,25]. Two studies by Abian-Vicen et al. [1] and Stojanović et al. [23] found higher levels of perceived exertion in the placebo than caffeine condition across several tests– RJ-15, CMJ, Yo-Yo IR-1, squat jump, LAD, a linear sprint of 5 m, 10 m, 20 m and suicide run.

Perceived muscle power was found to be higher following pre-exercise caffeine ingestion than placebo during a simulated basketball game protocol and a series of basketball-specific protocols [1,19]. Likewise, higher perceived endurance was found in both studies. However, the simulated basketball game by Puente et al. [19] reported only borderline minor increases in perceived endurance, which suggests that caffeine may not be ergogenic in improving one’s perception of their endurance performance during high-intensity bouts of exercise.

### 3.6. Genetic Variation

Recently, the importance of individual genetic variation has been highlighted due to the notion that some genes influence the ergogenic effect of caffeine on sports performance. The enzyme cytochrome P450 1A2 (CYP1A2) is a gene that is largely responsible for the speed of caffeine metabolism in the human body [9]. There are two types of caffeine metabolizers—C-allele carriers and AA homozygotes. The C-allele carriers are categorized as slow caffeine metabolizers while AA homozygotes are labeled as fast caffeine metabolizers. Only one study by Puente et al. [18] has investigated the influence of genetic variation on the ergogenic effects of caffeine for basketball performance (Table 5). The performance measures and test protocols used were previously reported in Puente et al. [19]. It was found that participants who were AA homozygotes performed significantly better in the Abalakov jump compared to the C-allele carriers (Table 5) after caffeine ingestion. It was also reported that the AA homozygotes had higher perceived muscle power as compared to their counterparts (Table 5). The metabolism of caffeine by CYP1A2 results in the formation of metabolites—paraxanthine, theobromine and theophylline. It was reported that paraxanthine and theophylline have a stronger capacity to bind with adenosine receptors [47]. The more rapid caffeine metabolism in the AA homozygotes could have resulted in a greater accumulation and production of paraxanthine and theophylline, enhancing the overall ergogenic effect on vertical jump and perceived muscle power [48]. Despite the findings in the mentioned study, there is insufficient evidence to conclude on the relationship of genetic variations with ergogenic outcomes from caffeine in basketball performance. Nevertheless, identification of genotype in relation to caffeine metabolism holds promise for more accurately determining the effects of caffeine on basketball.

### 3.7. Side Effects

Six studies in the current review measured the occurrence of side effects the day after ingesting caffeine [1,18,19,20,22,23]. Across the studies, a yes/no questionnaire containing 6 to 8-items was administered and these included: (1) headache; (2) abdominal/gastrointestinal discomfort; (3) muscle soreness/pain; (4) increased activeness; (5) tachycardia/heart palpitations; (6) insomnia; (7) irritability; (8) increased urine production/output; and (9) increased anxiety.

The most frequently occurring side effect was insomnia as some participants from all six studies reported experiencing insomnia after ingesting caffeine. The highest prevalence was noted by Puente and colleagues [18] where 70% of the AA homozygotes experienced insomnia (Table 6). Two other studies also reported a high prevalence of insomnia: caffeine vs. placebo = 54% vs. 19% [19] and 57% vs. 14% [20]. Conversely, the only trial that did not report any occurrence of insomnia was after completing an evening exercise protocol. This re-emphasizes the importance of identifying the genetic differences between individuals which could greatly affect the number or severity of side effects experienced by basketball players. Following a night training session or when having a match/tournament the next day, it is imperative for basketball players to obtain sufficient rest and recovery to prevent the loss of performance. Therefore, the use of caffeine by athletes should be closely monitored by appropriate professionals, such as team nutritionists or dieticians.

Increased activeness and vigor were other side effects that were evident, with the prevalence ranging from 9% [22] to as high as 37.5% [1]. Three of six studies (Table 6) reported at least 30% of the participants experiencing increased activeness (lower or no prevalence reported with placebo in the same studies) [1,18,23]. Two of these studies [18,23] involved female participants. It is possible that caffeine metabolism is linked with hormonal levels in the body. It was found that during the luteal phase of the menstrual cycle, estrogen and progesterone levels were greatly increased compared to the follicular phase [49]. As the female participants performed all tests during the luteal phase of their menstrual cycle, caffeine metabolism was suggested to be slower compared to other phases [2] as higher estrogen levels are associated with reduced caffeine metabolism [49]. This likely results in a rise in reported side effects. In addition, it was also noticed that urine output was increased following caffeine ingestion in studies that recorded this. Raya-González et al. [20] recorded 50% of their participants having increased urine production while Stojanović et al. [22] reported an increase in urine production from 45% of participants in the caffeine condition after the evening tests (Table 6).

A higher dose of caffeine could also result in greater side effects. Of the three studies that administered a moderate dose of 6 mg per kg of BM of caffeine [17,20,24], only Raya-González and colleagues [20] observed an increased number of side effects following caffeine ingestion. A 6 mg per kg of BM dose of caffeine was reported to result in greater occurrences of insomnia, tachycardia, abdominal discomfort, activeness, and urine output as compared to a placebo. Comparison of these data with lower doses of caffeine (3 mg per kg of BM) is difficult to determine because of the differences in protocol (e.g., the timing of caffeine ingestion) and participants involved but future work could examine this.

Collectively, there are many determinants that affect the occurrence of side effects in basketball players. However, it is also evident that pre-exercise caffeine ingestion still produces ergogenic effects on basketball performance. Hence, to ensure that basketball players fully capitalize on the ergogenic effect of caffeine, they should be mindful of their training and competition schedule as well as the individual genetic differences in metabolism before ingesting an appropriate dose. It is important that they experiment with caffeine ingestion during training to determine these effects before entering a competition period.

## 4. Conclusions

The present review focused on understanding the effects of caffeine on basketball performance, inclusive of skills (shooting and dribbling) and physical performance (vertical jump height, agility, linear and repeated sprints). Current evidence shows that pre-exercise caffeine ingestion, of a low or moderate dose, produces minimal to no ergogenic effect on basketball skills performance. Moreover, basketball skills performance did not deteriorate following caffeine ingestion. Conversely, caffeine has ergogenic effects on vertical jump, agility, and linear and repeated sprints. The ability to perform better in the physical aspects of play crucial to basketball will provide players with the upper hand during games. Overall, the use of caffeine in basketball will thus provide a boost to performance during competitions. However, a high prevalence of insomnia and increased activeness accompanies the consumption of caffeine, and this is influenced by individual genetic differences. Before utilizing it as an ergogenic substance, basketball players and coaches should seek to understand the properties of caffeine and properly plan for periods of ingestion to maximize its potential effects. Future research should seek to inform coaches and players of the effects of caffeine on basketball skill performance using protocols that are ecologically valid to the game for a better interpretation of the overall effects. Altogether, with better knowledge and understanding of the properties and effects of caffeine, basketball coaches, trainers, nutritionists, and players can utilize it effectively as an ergogenic substance to enhance overall basketball performance.

## Figures and Tables

**Figure 1 biology-11-00017-f001:**
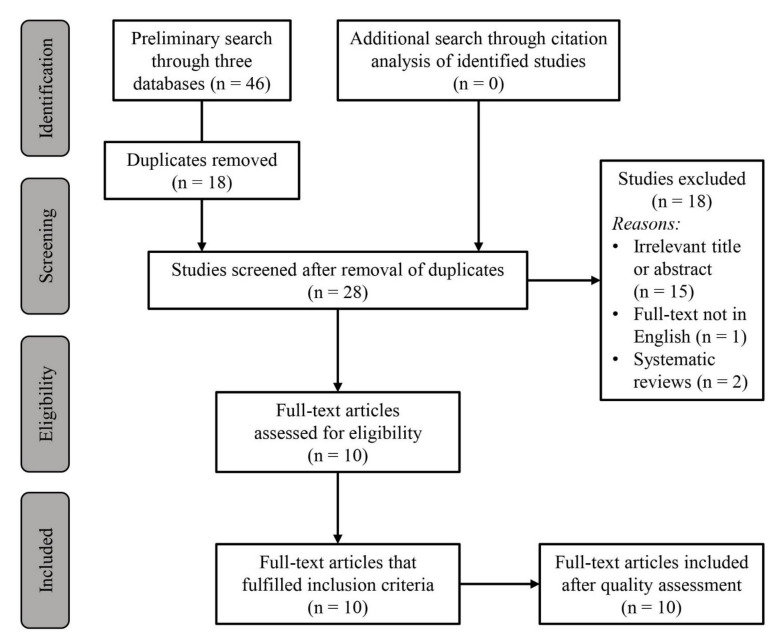
Search strategy and study selection process using PRISMA guidelines.

**Figure 2 biology-11-00017-f002:**
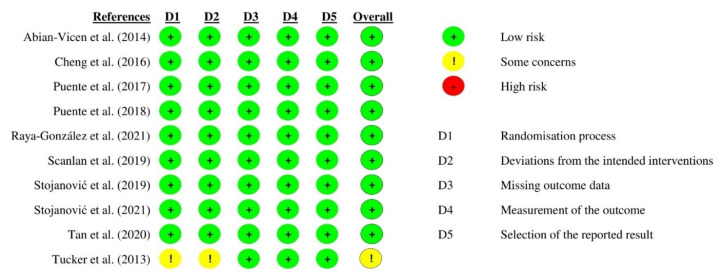
Cochrane Risk-of-Bias (RoB 2) tool.

**Table 1 biology-11-00017-t001:** Physiotherapy Evidence Database (PEDro) scale.

PEDro Criterion	Abian-Vicen et al., 2014 [1]	Cheng et al., 2016 [17]	Puente et al., 2018 [18]	Puente et al., 2017 [19]	Raya-Gonzalez et al., 2021 [20]	Scanlan et al., 2019 [21]	Stojanovic et al., 2021 [22]	Stojanovic et al., 2019 [23]	Tan et al., 2020 [24]	Tucker et al., 2013 [25]
1 *	0	0	1	1	1	0	0	0	1	0
2	1	1	1	1	1	1	1	1	1	0
3	1	1	1	1	1	1	1	1	0	0
4	0	0	0	0	0	0	0	0	0	0
5	1	1	1	1	1	1	1	1	1	1
6	1	1	1	1	1	1	1	1	0	1
7	1	1	1	1	1	1	1	1	0	1
8	1	1	1	1	1	1	1	1	1	1
9	1	1	1	1	1	1	1	1	1	1
10	1	1	1	1	1	1	1	1	1	1
11	1	1	1	1	1	1	1	1	1	0
Total	9/10	9/10	9/10	9/10	9/10	9/10	9/10	9/10	6/10	6/10

Criterion in the PEDro scale: 1= eligibility criteria; 2 = random allocation of subjects; 3 = concealed allocation; 4 = baseline comparability of important measures; 5 = blinding of subjects; 6 = blinding of therapists; 7 = blinding of assessors; 8 = measures obtained for >85% subjects; 9 = intention-to-treat analysis; 10 = between -group statistical comparisons; 11 = point measures and measures of variability. * Does not contribute to the total PEDro score. A score of ‘1’ indicates that the criterion is met while a score of ‘0′ indicates that the criterion is not met.

**Table 2 biology-11-00017-t002:** Effect of caffeine supplementation on basketball skills performance.

References	Participant Profile	Intervention	Administration Mode	Ingestion Time (min)	Measures (Units) & Change	CAF	PLA	*p*	Hedge’s *g* ES
Abian-Vicen et al., 2014	16 males [age: 14.9 ± 0.8 years; body mass = 73.4 ± 12.4 kg; height = 182.3 ± 6.5 cm]; National Spanish League first division junior team	3 mg per kg BM CAF or PLA; double-blind, counterbalanced, randomized design	Energy drink powder dissolved in 250 mL water	60	Free-throw (%): –	70.3 ± 11.8	70.7 ± 11.8	0.45	−0.03
					3-point score (%): –	39.9 ± 11.8	38.1 ± 12.8	0.33	0.15
Puente et al., 2017	10 males [age: 27.1 ± 4.0 years] and 10 females [age: 27.9 ± 6.1 years]; professional and semi-professional level	3 mg per kg BM CAF or PLA; double-blind, counterbalanced, randomized design	Capsule	60	Free-throw score during				
					basketball-specific testing: –	15.6 ± 2.3	15.4 ± 1.6	0.39	0.10
					CODAT (s):				
					With the ball: –	6.14 ± 0.32	6.20 ± 0.29	0.12	−0.20
					Notational Analysis –				
					2-point field goals made: –	2.7 ± 2.6	2.5 ± 2.4	0.37	0.08
					2-point field goals attempted: –	4.5 ± 3.3	3.8 ± 3.0	0.21	0.22
					Accuracy in 2-point field goals (%): –	52.9 ± 37.2	54.7 ± 30.5	0.45	−0.05
					3-point field goals made: –	0.8 ± 1.1	0.9 ± 1.2	0.27	−0.09
					3-point field goals attempted: –	2.4 ± 2.3	2.8 ± 2.1	0.23	−0.18
					Accuracy in 3-point field goals (%): –	23.7 ± 27.5	27.4 ± 31.5	0.33	−0.13
					Free-throws made: ↑ *	1.1 ± 1.1	0.6 ± 0.8	0.03	0.53
					Free-throws attempted: ↑ *	1.5 ± 1.5	0.9 ± 1.1	0.04	0.46
					Accuracy in free-throws (%): –	73.8 ± 20.7	71.4 ± 40.5	0.44	0.08
Scanlan et al., 2019	11 males and 10 females [age = 18.3 ± 3.3 years; body mass = 72.6 ± 7.5 kg; height = 180.3 ± 7.2 cm]; elite level	3 mg per kg BM CAF or PLA; double-blind, counterbalanced; randomized, crossover design	Capsule with 250 mL water	60	Total dribble time (s):				
					5 m: –	1.16 ± 0.08	1.17 ± 0.10	0.34	−0.14
					10 m: –	2.00 ± 0.13	2.01 ± 0.13	0.64	−0.08
					20 m: –	3.53 ± 0.21	3.56 ± 0.18	0.33	−0.16
					Dribble deficit (s):				
					5 m: –	0.03 ± 0.10	0.01 ± 0.11	0.21	0.20
					10 m: –	0.07 ± 0.09	0.03 ± 0.10	0.17	0.42
					20 m: –	0.15 ± 0.13	0.15 ± 0.14	0.89	0.04
Stojanović et al., 2019	10 females [age: 20.2 ± 3.9 years; body mass: 69.2 ± 6.3 kg; height 175.4 ± 5.9 cm]; professional level	3 mg per kg BM CAF or PLA; double-blind, counterbalanced, randomized design	Capsule with 250 mL water	60	Linear sprint time (s):				
					5 m dribbling sprint: –	1.20 ± 0.05	1.22 ± 0.08	0.45	−0.31
					10 m dribbling sprint: –	2.05 ± 0.12	2.07 ± 0.11	0.55	−0.17
					20 m dribbling sprint: –	3.56 ± 0.25	3.65 ± 0.15	0.15	−0.45
Stojanović et al., 2021	11 males [age: 16.5 ± 1.0 years; body mass: 75.7 ± 7.4 kg; height: 184.7 ± 5.0 cm]; national youth level	3 mg per kg BM CAF or PLA; double-blind, counterbalanced, randomized design	Capsule with 250 mL water	60	Morning				
					Repeated-sprint performance (s):				
					Suicide run with dribbling time: –	28.50 ± 2.06	29.04 ± 2.25	0.62	−0.23
					Evening				
					Repeated-sprint performance (s):				
					Suicide run with dribbling time: –	32.65 ± 2.01	32.35 ± 1.93	0.62	0.14
Tan et al., 2020	12 males [age: 23.1 ± 1.9 years; body mass: 77.1 ± 12.4 kg; height: 180.1 ± 8.8 cm] and six females [age: 22.0 ± 1.3 yr; body mass: 67.0 ± 11.1 kg; height: 169.4 ± 8.9 cm]; college level	6 mg per kg BM CAF or PLA; single-blind, randomized design	Caffeine powder dissolved in 300 mL water	60	Free-throw scores: –	6.1 ± 1.7	5.5 ± 2.0	0.34	0.32

Data are reported as mean ± SD. CAF caffeine, PLA placebo, BM body mass, ES effect size, CODAT change of direction and acceleration test, – no change, ↑ increased, * significant change.

**Table 3 biology-11-00017-t003:** Effect of caffeine supplementation on physical performances of basketball players.

References	Participant Profile	Intervention	Administration Mode	Ingestion Time (min)	Measures (Units) & Change	CAF	PLA	*p*	Hedge’s *g* ES
Abian-Vicen et al., 2014	16 males [age: 14.9 ± 0.8 years; body mass = 73.4 ± 12.4 kg; height = 182.3 ± 6.5 cm]; National Spanish League first division junior team	3 mg per kg BM CAF or PLA; double-blind, counterbalanced, randomized design	Energy drink powder dissolved in 250 mL water	60	CMJ without arm swing (cm): ↑ *	38.3 ± 4.4	37.5 ± 4.4	<0.05	0.18
					Mean power output (W/kg): –	30.4 ± 2.8	30.1 ± 3.5	0.32	0.10
					Peak power output (W/kg): –	53.9 ± 5.0	53.8 ± 5.5	0.45	0.02
					RJ-15 (cm): ↑ *	30.2 ± 3.6	28.8 ± 3.4	<0.05	0.40
					Leg muscle power output (W/kg): ↑ *	51.4 ± 5.7	49.4 ± 4.6	<0.05	0.39
					Yo-Yo IR-1 test (m): –	2000 ± 706	1925 ± 702	0.19	0.11
Cheng et al., 2016	15 males [age: 20 ± 2 years; body mass: 84 ± 12 kg; 188 ± 6 cm]; Division I college level	6 mg per kg BM CAF or PLA; double-blind, randomized, crossover design	Capsule with 200 mL water	60	VO2peak: –	48.0 ± 7.5	49.2 ± 6.3	>0.05	−0.17
					EP (W): –	242 ± 37	244 ± 42	NS	−0.05
					WEP (kJ): ↑ *	13.4 ± 3.0	12.1 ± 2.7	<0.05	0.46
					Peak power (W): –	538 ± 60	537 ± 49	NS	0.02
					Mean power (W): –	316 ± 34	311 ± 38	NS	0.14
					Total work (kJ):	56.9 ± 6.2	56.0 ± 6.9	NS	0.14
					PO30 (W): –	472 ± 45	471 ± 43	NS	0.02
					PO60 (W): ↑ *	419 ± 37	410 ± 38	<0.05	0.24
					PO90 (W): ↑ *	377 ± 35	368 ± 37	<0.05	0.25
					PO120 (W): ↑ *	349 ± 34	341 ± 38	<0.05	0.22
					PO150 (W): –	330 ± 35	324 ± 38	0.09	0.16
					FR (/s): ↑ *	0.024 ± 0.007	0.029 ± 0.006	0.01	−1.23
Puente et al., 2017	10 males [age: 27.1 ± 4.0 years] and 10 females [age: 27.9 ± 6.1 years]; professional and semi-professional level	3 mg per kg BM CAF or PLA; double-blind, counterbalanced, randomized design	Capsule	60	Abalakov jump (cm): ↑ *	38.2 ± 7.4	37.3 ± 6.8	0.012	0.13
					CODAT (s):				
					Without the ball: –	5.95 ± 0.31	5.96 ± 0.29	0.388	−0.03
Raya-González et al., 2021	14 males [age: 21 ± 2 years; body mass: 87 ± 6 kg; height 190 ± 5 cm]; professional level	6 mg per kg BM CAF or PLA; double-blind, counterbalanced, randomized, crossover design	Supplement dissolved in 250 mL water	60	Fitness tests:				
					CMJ without arm swing (cm): ↑ *	39.49 ± 5.28	37.09 ± 5.14	0.02	0.46
					20 m sprint (s): ↑ *	3.05 ± 0.15	3.22 ± 0.15	<0.001	−1.13
					Lane Agility Drill time (s): ↑ *	11.56 ± 0.57	12.13 ± 0.82	<0.001	−0.82
					Total RSA performance (s): ↑ *	33.53 ± 1.33	35.23 ± 1.62	<0.001	−1.15
					Best individual RSA (s): ↑ *	5.50 ± 0.22	5.75 ± 0.27	<0.001	−1.02
Stojanović et al., 2019	10 females [age: 20.2 ± 3.9 years; body mass: 69.2 ± 6.3 kg; height 175.4 ± 5.9 cm]; professional level	3 mg per kg BM CAF or PLA; double-blind, counterbalanced, randomized design	Capsule with 250 mL water	60	Jump height (cm):				
					CMJ without arm swing: –	29.20 ± 4.39	27.92 ± 4.24	0.10	0.30
					CMJ with arm swing: –	35.14 ± 5.08	33.85 ± 3.92	0.15	0.29
					Squat jump: –	27.22 ± 4.37	25.97 ± 3.16	0.08	0.33
					Change-of-direction time (s):				
					Lane Agility Drill time: –	12.99 ± 0.86	13.22 ± 0.87	0.12	−0.27
					Linear sprint time (s):				
					5 m sprint: –	1.18 ± 0.11	1.24 ± 0.15	0.13	−0.46
					10 m sprint: ↑ *	2.01 ± 0.13	2.11 ± 0.18	0.05	−0.65
					20 m sprint: ↑ *	3.49 ± 0.23	3.59 ± 0.25	0.04	−0.42
					Repeated sprint performance (s):				
					Suicide run time: –	31.80 ± 1.62	32.20 ± 1.74	0.28	−0.24
Stojanović et al., 2021	11 males [age: 16.5 ± 1.0 years; body mass: 75.7 ± 7.4 kg; height: 184.7 ± 5.0 cm]; national youth level	3 mg per kg BM CAF or PLA; double-blind, counterbalanced, randomized design	Capsule with 250 mL water	60	Morning				
					Jump height (cm):				
					CMJ without arm swing: ↑ *	33.90 ± 5.38	31.03 ± 4.98	<0.001	0.51
					CMJ with arm swing: ↑ *	42.32 ± 5.69	39.98 ± 5.23	<0.001	0.40
					Squat jump: ↑ *	33.20 ± 4.71	30.55 ± 4.89	<0.001	0.51
					Change-of-direction speed (s):				
					Lane Agility Drill time: ↑ *	11.98 ± 0.70	12.46 ± 0.75	<0.05	−0.61
					Linear sprint time (s):				
					5 m sprint time: –	1.15 ± 0.11	1.16 ± 0.08	>0.05	−0.09
					10 m sprint time: –	1.95 ± 0.14	1.95 ± 0.12	>0.05	0.00
					20 m sprint time: –	3.33 ± 0.22	3.38 ± 0.21	>0.05	−0.21
					Repeated-sprint performance (s):				
					Suicide run time: ↑ *	26.49 ± 1.62	27.26 ± 1.52	<0.001	−0.45
					Evening				
					Jump height (cm):				
					CMJ without arm swing: –	33.92 ± 6.05	33.58 ± 5.84	>0.05	0.05
					CMJ with arm swing: –	42.23 ± 6.06	42.89 ± 6.04	>0.05	−0.10
					Squat jump: –	32.27 ± 5.06	31.74 ± 6.42	>0.05	0.08
					Change-of-direction speed (s):				
					Lane Agility Drill time: –	12.61 ± 0.84	12.59 ± 0.87	>0.05	0.02
					Linear sprint time (s):				
					5 m sprint time: –	1.08 ± 0.08	1.09 ± 0.08	>0.05	−0.11
					10 m sprint time: –	1.86 ± 0.12	1.87 ± 0.09	>0.05	−0.08
					20 m sprint time: –	3.27 ± 0.18	3.25 ± 0.17	>0.05	0.10
					Repeated-sprint performance (s):				
					Suicide run time: –	29.91 ± 1.31	29.97 ± 1.63	>0.05	−0.04
Tucker et al., 2013	5 males [age: 22 ± 1.6 years; body mass: 84.6 ± 8.3 kg; height: 187.4 ± 7.9 cm]; elite level	3 mg per kg BM CAF or PLA; double-blind, counterbalanced, randomized design	Tablet containing B1 thiamine	60	Reactive strength index (cm/s):				
					Participant A: –	124 ± 5	118 ± 13	0.081	0.67
					Participant B: –	117 ± 13	126 ± 16	0.161	−0.62
					Participant C: ↑ *	119 ± 9	109 ± 6	0.013	1.33
					Participant D: –	111 ± 9	122 ± 14	0.081	−0.96
					Participant E: –	87 ± 6	83 ± 4	0.154	0.80

Data are reported as mean ± SD. CAF caffeine, PLA placebo, BM body mass, ES effect size, CMJ countermovement jump, RJ-15 15 s maximal jump test, Yo-Yo IR-1 Yo-Yo Intermittent Recovery Level-1, VO2peak peak oxygen uptake, EP end-test power, WEP work done above EP, PO30 averaged power output from 0 to 30 s, PO60 averaged power output from 0 to 60 s, PO90 averaged power output from 0 to 90 s, PO120 averaged power output from 0 to 120 s, PO150 averaged power output from 0 to 150 s, FR fatigue rate, CODAT change of direction and acceleration test, RSA repeated sprint ability, – no change, ↑ increased, * significant change.

**Table 4 biology-11-00017-t004:** Effect of caffeine on physiological responses of basketball players.

References	Participant Profile	Intervention	Administration Mode	Ingestion Time (min)	Measures (Units) & Change	CAF	PLA	*p*	Hedge’s *g* ES
Abian-Vicen et al., 2014	16 males [age: 14.9 ± 0.8 years; body mass = 73.4 ± 12.4 kg; height = 182.3 ± 6.5 cm]; National Spanish League first division junior team	3 mg per kg BM CAF or PLA; double-blind, counterbalanced, randomized design	Energy drink powder dissolved in 250 mL water	60	Perceived muscle power: ↑ *	7.1 ± 1.1	5.2 ± 1.2	<0.05	1.65
					Perceived endurance: ↑ *	6.6 ± 1.4	5.1 ± 1.1	<0.05	1.20
					Perceived exertion: ↑ *	4.6 ± 1.8	5.7 ± 2.3	<0.05	0.44
Cheng et al., 2016	15 males [age: 20 ± 2 years; body mass: 84 ± 12 kg; 188 ± 6 cm]; Division I college level	6 mg per kg BM CAF or PLA; double-blind, randomized, crossover design	Capsule with 200 mL water	60	Peak heart rate (bpm): ↑ *	172 ± 7	165 ± 8	<0.05	0.93
					Perceived exertion: –	18.7 ± 1.5	18.7 ± 1.0	>0.05	0.00
					Blood lactate (mmol/L): ↑ *	11.25 ± 2.58	9.80 ± 1.76	<0.05	0.67
Puente et al., 2017	10 males [age: 27.1 ± 4.0 years] and 10 females [age: 27.9 ± 6.1 years]; professional and semi-professional level	3 mg per kg BM CAF or PLA; double-blind, counterbalanced, randomized design	Capsule	60	Heart rate (bpm):				
					Mean: –	161 ± 10	157 ± 13	0.229	0.35
					Maximal: –	188 ± 10	185 ± 12	0.499	0.27
					Perceived exertion (A.U.): –	4.9 ± 1.5	5.3 ± 1.6	0.396	−0.26
					Perceived muscle power (A.U.): ↑ *	6.6 ± 1.4	5.3 ± 1.4	0.003	0.50
					Perceived endurance (A.U.): –	6.3 ± 1.6	5.5 ± 1.2	0.058	0.57
Stojanović et al., 2019	10 females [age: 20.2 ± 3.9 years; body mass: 69.2 ± 6.3 kg; height 175.4 ± 5.9 cm]; professional level	3 mg per kg BM CAF or PLA; double-blind, counterbalanced, randomized design	Capsule with 250 mL water	60	Perceived exertion (AU): ↑ *	5.6 ± 2.5	7.8 ± 1.2	0.04	−1.19
					Perceived performance (AU): –	4.2 ± 2.7	3.6 ± 2.8	0.53	0.22
Stojanović et al., 2021	11 males [age: 16.5 ± 1.0 years; body mass: 75.7 ± 7.4 kg; height: 184.7 ± 5.0 cm]; national youth level	3 mg per kg BM CAF or PLA; double-blind, counterbalanced, randomized design	Capsule with 250 mL water	60	Morning				
					Perceived exertion (AU): –	5.4 ± 1.3	6.1 ± 1.8	>0.05	−0.45
					Perceived performance (AU): –	6.2 ± 2.6	5.7 ± 2.2	>0.05	0.21
					Tympanic temperature (°C): ↑ *	36.2 ± 0.3	36.1 ± 0.4	0.04	0.29
					Evening				
					Perceived exertion (AU): –	7.3 ± 1.6	7.3 ± 1.5	>0.05	0.00
					Perceived performance (AU): –	4.9 ± 2.0	5.7 ± 2.5	>0.05	−0.36
					Tympanic temperature (°C): ↑ *	36.6 ± 0.3	36.6 ± 0.4	0.04	0.00
Tan et al., 2020	12 males [age: 23.1 ± 1.9 years; body mass: 77.1 ± 12.4 kg; height: 180.1 ± 8.8 cm] and six females [age: 22.0 ± 1.3 yr; body mass: 67.0 ± 11.1 kg; height: 169.4 ± 8.9 cm]; college level	6 mg per kg BM CAF or PLA; single-blind, randomized design	Caffeine powder dissolved in 300 mL water	60	Heart rate (bpm):				
					Set 1: ↑ *	159 ± 12.2	154 ± 15.6		0.36
					Set 2: ↑ *	160 ± 8.7	154 ± 10.7		0.62
					Set 3: ↑ *	163 ± 9.7	158 ± 11.7	0.02	0.49
					Set 4: –	162 ± 11.9	161 ± 9.4		0.09
					Set 5: ↑ *	166 ± 9.2	163 ± 12.1		0.28
					Rate of perceived exertion:				
					Set 1: –	10.9 ± 2.3	11.7 ± 3.0		−0.30
					Set 2: –	12.6 ± 2.0	12.5 ± 2.5		0.04
					Set 3: –	13.7 ± 2.1	13.4 ± 2.2	0.57	0.14
					Set 4: –	14.6 ± 1.7	14.7 ± 1.9		−0.06
					Set 5: –	15.8 ± 2.1	15.7 ± 1.9		0.05
Tucker et al., 2013	5 males [age: 22 ± 1.6 years; body mass: 84.6 ± 8.3 kg; height: 187.4 ± 7.9 cm]; elite level	3 mg per kg BM CAF or PLA; double-blind, counterbalanced, randomized design	Tablet containing B1 thiamine	60	Blood lactate (mmol/L): ↑	NS	NS	NS	NS
					Heart rate (bpm): NS	NS	NS	NS	NS
					Rate of Perceived Exertion:				
					Overall: –	12.8 ± 4.0	12.8 ± 4.4	0.125	0.00
					Legs: –	12.9 ± 3.9	12.8 ± 4.3	0.406	0.02
					Respiratory Exchange Ratio:				
					Participant A: ↑	NS	NS	<0.001	NS
					Participant B: ↑	NS	NS	<0.001	NS
					Participant C: –	NS	NS	0.75	NS
					Participant D: ↑	NS	NS	<0.001	NS
					Participant E: –	NS	NS	0.58	NS

Data are reported as mean ± SD. CAF caffeine, PLA placebo, BM body mass, ES effect size, bpm beats per minute, A.U. arbitrary units, °C degree Celsius, NS not specified, – no change, ↑ increased, * significant change.

**Table 5 biology-11-00017-t005:** Effect of caffeine on physical and basketball performance across different genetic types.

References	Participant Profile	Intervention	Administration Mode	Ingestion Time (min)	Measures (Units) & Change	CAF	PLA	*p*	Hedge’s *g* ES
Puente et al., 2018	10 AA homozygotes [age: 26.7 ± 3.5 years; body mass: 83.5 ± 19.2 kg; height: 187.6 ± 16.7 cm] and 9 C-allele carriers [age: 29.4 ± 6.0 yr; body mass: 78.4 ± 14.7 kg; height: 182.8 ± 16.7 cm]; professional and semi-professional level	3 mg per kg BM CAF or PLA; double-blind, randomized design	Capsule	60	Mean Abalakov jump height (cm):				
					AA homozygotes: ↑ *	40.7 ± 7.3	39.6 ± 7.2	0.03	0.15
					C-allele carriers: –	37.2 ± 6.9	36.3 ± 5.9	0.33	0.14
					CODAT without the ball (s):				
					AA homozygotes: –	5.88 ± 0.27	5.91 ± 0.25	0.36	−0.12
					C-allele carriers: –	5.97 ± 0.38	5.95 ± 0.33	0.37	0.06
					CODAT with the ball (s):				
					AA homozygotes: –	6.09 ± 0.24	6.19 ± 0.21	0.15	−0.44
					C-allele carriers: –	6.14 ± 0.41	6.14 ± 0.35	0.49	0.00
					Mean heart rate (bpm):				
					AA homozygotes: –	160 ± 10	158 ± 9	0.72	0.21
					C-allele carriers: –	163 ± 9	161 ± 13	0.82	0.18
					Peak heart rate (bpm):				
					AA homozygotes: –	188 ± 13	187 ± 12	0.22	0.08
					C-allele carriers: –	185 ± 6	182 ± 7	0.46	0.23
					Perceived muscle power (A.U.):				
					AA homozygotes: ↑ *	6.7 ± 1.3	5.3 ± 1.8	0.04	0.90
					C-allele carriers: –	6.2 ± 1.5	5.4 ± 0.9	0.16	0.67
					Perceived exertion (A.U.):				
					AA homozygotes: –	4.6 ± 1.5	5.3 ± 1.6	0.20	−0.45
					C-allele carriers: –	5.4 ± 1.5	5.4 ± 1.5	0.50	0.00
					Perceived endurance (A.U.):				
					AA homozygotes: –	6.8 ± 1.5	5.7 ± 1.6	0.06	0.71
					C-allele carriers: –	5.6 ± 1.7	5.6 ± 0.9	0.50	0.00

Data are reported as mean ± SD. CAF caffeine, PLA placebo, BM body mass, ES effect size, bpm beats per minute, A.U. arbitrary units, —no change, ↑ increased, * significant change.

**Table 6 biology-11-00017-t006:** Side effects of caffeine during basketball testing studies.

References	Participant Profile	Intervention	Administration Mode	Ingestion Time (min)	Side Effects	CAF (%)	PLA (%)
Abian-Vicen et al., 2014	16 males [age: 14.9 ± 0.8 years; body mass = 73.4 ± 12.4 kg; height = 182.3 ± 6.5 cm]; National Spanish League first division junior team	3 mg per kg BM CAF or PLA; double-blind, counterbalanced, randomized design	Energy drink powder dissolved in 250 mL water	60	Headache:	6.3	12.5
					Abdominal discomfort:	12.5	6.3
					Muscle soreness:	31.3	25
					Increased activeness:	37.5	0
					Tachycardia and heart palpitations:	0	0
					Insomnia:	12.5	0
					Increased urine production:	0	0
					Increased anxiety:	0	0
Puente et al., 2017	10 males [age: 27.1 ± 4.0 years] and 10 females [age: 27.9 ± 6.1 years]; professional and semi-professional level	3 mg per kg BM CAF or PLA; double-blind, counterbalanced, randomized design	Capsule	60	Insomnia:	54.4	19.0
					Nervousness:	NS	NS
					Irritability:	NS	NS
					Activeness:	NS	NS
					Gastrointestinal discomfort:	NS	NS
					Headache:	NS	NS
					Muscle pain:	NS	NS
Puente et al., 2018	10 AA homozygotes [age: 26.7 ± 3.5 years; body mass: 83.5 ± 19.2 kg; height: 187.6 ± 16.7 cm] and 9 C-allele carriers [age: 29.4 ± 6.0 yr; body mass: 78.4 ± 14.7 kg; height: 182.8 ± 16.7 cm]; professional and semi-professional level	3 mg per kg BM CAF or PLA; double-blind, randomized design	Capsule	60	Nervousness:		
					AA homozygotes	20	10
					C-allele carriers:	0	11
					Insomnia:		
					AA homozygotes	70	20
					C-allele carriers:	33	22
					Gastrointestinal complaints:		
					AA homozygotes	20	0
					C-allele carriers:	0	0
					Activeness:		
					AA homozygotes	30	20
					C-allele carriers:	11	11
					Muscle pain:		
					AA homozygotes	10	30
					C-allele carriers:	11	11
					Headache:		
					AA homozygotes	0	10
					C-allele carriers:	0	11
Raya-González et al., 2021	14 males [age: 21 ± 2 years; body mass: 87 ± 6 kg; height 190 ± 5 cm]; professional level	6 mg per kg BM CAF or PLA; double-blind, counterbalanced, randomized, crossover design	Supplement dissolved in 250 mL water	60	Insomnia:	57	14
					Tachycardia:	14	0
					Anxiety:	0	0
					Abdominal discomfort:	21	14
					Headache:	0	7
					Activeness:	21	7
					Muscle soreness:	7	14
					Urine output:	50	7
Stojanović et al., 2019	10 females [age: 20.2 ± 3.9 years; body mass: 69.2 ± 6.3 kg; height 175.4 ± 5.9 cm]; professional level	3 mg per kg BM CAF or PLA; double-blind, counterbalanced, randomized design	Capsule with 250 mL water	60	Headache:	10	20
					Abdominal discomfort:	20	10
					Muscle soreness:	10	0
					Increased activeness:	30	0
					Tachycardia:	30	10
					Insomnia:	10	20
					Increased urine production:	10	10
					Increased anxiety:	0	10
Stojanović et al., 2021	11 males [age: 16.5 ± 1.0 years; body mass: 75.7 ± 7.4 kg; height: 184.7 ± 5.0 cm]; national youth level	3 mg per kg BM CAF or PLA; double-blind, counterbalanced, randomized design	Capsule with 250 mL water	60	Morning	9	0
					Headache:	0	0
					Abdominal discomfort:	9	0
					Muscle soreness:	18	18
					Increased activeness:	0	0
					Tachycardia:	9	9
					Insomnia:		
					Increased urine production:	9	0
					Increased anxiety:	0	0
					Evening	9	9
					Headache:	18	9
					Abdominal discomfort:	18	9
					Muscle soreness:	9	27
					Increased activeness:	27	9
					Tachycardia:	0	0
					Insomnia:		
					Increased urine production:	45	9
					Increased anxiety:	0	9

CAF caffeine, PLA placebo, BM body mass.

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
