# Peer review of "A Systematic Review of the Effects of Caffeine on Basketball Performance Outcomes"

_biology, 2021, doi:10.3390/biology11010017_

Round 1

Reviewer 1 Report

The present systematic review set out to determine the impact caffeine consumption has on markers of basketball performance. The overall paper is written exceptionally well and thus, I have almost no feedback on the actual writing itself. There are a couple of grammatical errors that will improve this paper though. Regarding methodology, I have no major concerns. Following these minor changes, my opinion is that this paper will contribute nicely to the body of literature surrounding caffeine supplementation and basketball performance.

Line 70 - Effects should read affects

Line 202 - 6.0 should just read 6, like the other dosages where decimals were not used. Consistency is key here

Lines 211, 214, and 248- in some places, the authors state mg per kg and then here, they state mg/kg. Again, for consistency purposes, change these to one format and keep the same throughout.

Line 310 - Technically, the space between 140 and m is correct. Above though (Line 304) it reads 5m,10m, and 20m. Either create a space between the number and unit or close the space, but keep consistent.

Overall, well written and formulated paper.

Reviewer 2 Report

Comments and Suggestions for Authors: 

 This review paper selected 10 research articles from the PubMed data base focusing on the effects of caffeine on basketball performance which includes basketball skills performance (shooting accuracy and dribbling speed), physical performance (vertical jump, agility, linear and repeated sprints, and power output), physiological and subjective responses (heart rate, blood lactate, and self-perceived responses), and side effects. The authors indicated that pre-exercise caffeine ingestion produces minimal to no ergogenic effect on basketball skills performance. Conversely, the authors claimed that caffeine has ergogenic effects on physical performance, such as vertical jump, agility, and linear and repeated sprints. The research also shows that high prevalence of insomnia and increased activeness accompanies the consumption of caffeine, and this is influenced by individual genetic differences.  

 In general, the authors discussed the data from 10 articles and summarized the effects of caffeine on basketball players, but the conclusions here are also not novel and the authors did not provide any promising outlooks. 

 General concept comments  

 The literature search and selection strategy: I searched the key word “caffeine & basketball” against the data base of PubMed which yields 12 results: 10 ten of them contain words of “caffeine & basketball” in the titles are the ones selected in this review, suggesting the method used in this review is too complicated and unnecessary. 

 Overall, caffeine may only impact physical performance, not skills performance. However, physical performance not only refers to the basketball player, but it also influences all other sports fields. As this review only collected the data from 130 participants, which is not convincing enough, research with regarding to athletes from other sport fields and public people should also be summarized to evaluate the impact of caffeine on the physical performance. This review could be more comprehensive if the authors had included more data of physical performances by caffeine from not only basketball players, as the number of the participants is little here. 

 The authors analyzed the results from 10 articles selected and attempted to compare the effects of caffeine between males and females, while current data is not sufficient to identify the difference. The data in the tables is not merged or analyzed systematically, such as by gender or ages. 

 The authors should double check the data from the articles or merge the data and redo the data analysis. 

 Specific comments: 

 Line 155-157: “Of the studies reviewed, the common placebos used were cellulose [17–19], dextrose [21–23], sucrose [20], maltodextrin [24], zero caffeine containing energy drink [1], and vitamin tablets [25].” In these studies, whether the placebos used would also influence performance (why not just use water as placebos and water with caffeine as the stimulant); why a positive control (another stimulant) was not tested in parallel. 

Table 3 summarizes the effect of caffeine supplementation on physical performances of basketball players, including jump height, and sprints. The data here is not comprehensive. Table 3 lists the results from refs 1, 17, 19, 20, 22, 23, and 25, among which refs 1, 19, 20, 22, and 23 include the test of jumping height and refs 20, 22, and 23 tested sprinting. However, reference 18 also includes the Abalakov jump test, and reference 25 performed 10 vertical rebound jumps. Refs 23 and 25 show negative results in jumping height test. Sprinting is also tested in refs 18, and 21 and the results from refs 18, and 22 show negative results.  

Table 3: The data from ref 22 (Stojanović et al., 2021) shows negative results in sprint time, which is conflictive to the original description in ref 22 like “PMPLAC demonstrated small-moderate, significant (p<0.05) improvements in CMJ (ES = 0.43), CMJAS (ES = 0.48), and 20-m sprint (ES = -0.63) compared to AMPLAC” and “AMCAFF produced small-moderate improvements in vertical jump, change-of-direction, 20-m linear sprint, and repeated-sprint performance compared to AMPLAC”. 

Table 3: The p value in this table is also abnormal, such as the “CMJ with arm swing” in the evening from ref 22 (Stojanović et al., 2021) in page 17: CAF=42.23 ± 6.06, PLA= 42.89 ± 6.04, and the p <0.001 which indicates significant difference. Similar data like this can be found in this table. 

 Line 149: Four studies included female athletes should be references 18, 21, 23, and 24. Please double check it. 

Round 2

Reviewer 2 Report

The authors addressed most of the questions.

Here are some minor comments:

Line 644: utilise to utilize

Appendix B

Tabel B1, the data could be further analyzed to indicate the effect of caffeine as no change, increased, or significant change.

Author Response

We would like to thank the Reviewer for the further comments on our work. We have tried to address these to the best of our ability and believe that the revised version of the manuscript is improved. Please refer to the responses provided below.

From Reviewer 2:

  1. Line 644: utilise to utilize. 

Response: Change made as requested. 

  1. Table B1, the data could be further analyzed to indicate the effect of caffeine as no change, increased, or significant change. 

Response: Thank you for the comment. We have indicated in Table B1 (Appendix) using arrows if there was a change in each parameter measured for men and women.